# Understanding the Role of NLRP3 Inflammasome in Acute Pancreatitis

**DOI:** 10.3390/biology13110945

**Published:** 2024-11-18

**Authors:** Konstantinos Papantoniou, Ioanna Aggeletopoulou, Christos Michailides, Ploutarchos Pastras, Christos Triantos

**Affiliations:** 1Department of Internal Medicine, University Hospital of Patras, 26504 Patras, Greece; g.papanton@yahoo.gr (K.P.); christos.mich1@gmail.com (C.M.); 2Division of Gastroenterology, Department of Internal Medicine, University Hospital of Patras, 26504 Patras, Greece; iaggel@upatras.gr (I.A.); ploutarchosp96@gmail.com (P.P.)

**Keywords:** NLRP3 inflammasome, activation, pancreatitis, prognosis, inhibitors

## Abstract

Acute pancreatitis (AP) is a serious disease of the pancreas, which can cause many complications, including multi-organ dysfunction and death. Treatment is mainly supportive, making research for the discovery of new therapeutic agents for treating AP essential. Inflammasomes, including the NOD-like receptor family pyrin domain-containing 3 (NLRP3) inflammasome, are multi-protein complexes involved in many inflammatory processes, such as the one that promotes tissue damage in patients with AP. This makes NLRP3 a possible target for AP-directed treatment. Despite promising results from animal studies, few clinical trials have examined the possible use of NLRP3 inhibitors in treating AP patients. Future research focusing on the effect of NLRP3 inhibitors in human patients with AP, as well as possible adverse events associated with their use, can help improve patient care.

## 1. Introduction

Acute pancreatitis (AP) is a serious condition, characterized by severe pancreatic inflammation and damage. Treatment for AP remains mainly supportive, and measures such as fluid resuscitation, pain management, early oral refeeding and early recognition of organ dysfunction are a mainstay in clinical practice. However, these measures mainly focus on managing symptoms and preventing complications. Due to lack of a direct effect on the underlying inflammatory process, current AP treatment does not always cease progression of the disease, with many patients developing severe complications, including multi-organ dysfunction and death [1]. Pharmacological therapies targeting the pathogenesis of AP are currently lacking. The effects of several agents including antibiotics, antioxidants, non-steroidal anti-inflammatory drugs (NSAIDs) and probiotics have been examined in clinical trials in patients with AP, but none of them exhibited a consistent positive effect and reduction in AP complications, including mortality [2]. Inflammasomes are a group of multi-protein complexes that participate in innate immunity processes through the activation of pro-inflammatory caspases [3]. Their involvement in the pathogenesis of many digestive diseases, including AP [4,5], makes their inhibition a reasonable therapeutic target.

Nucleotide oligomerization domain (NOD)-like receptor family pyrin domain-containing 3 (NLRP3) contributes to the inflammatory process in many different conditions, including Alzheimer’s disease and atherosclerosis [6,7]. The aim of this review is to explore the importance of NLRP3 activation and its mechanistic contributions to the inflammatory process in AP. We delve into how NLRP3 can potentially serve as a valuable prognostic biomarker in AP patients, offering new insights into disease progression and outcomes. Additionally, we highlight emerging NLRP3 inhibitors that show promise for transforming the clinical management of AP, presenting them as potential future therapeutic options that could significantly enhance patient care and treatment strategies. Through this comprehensive overview, we aim to shed light on innovative approaches that could reshape how AP is prognosed and treated.

## 2. The Innate Immune System and Inflammasomes

Unlike the adaptive immune system, which develops a targeted response over time, the innate immune system provides a rapid, non-specific response to a wide variety of stimuli. Innate immune cells include many types of cells, such as neutrophils and peripheral blood mononuclear cells (PBMCs). Interaction with a variety of molecules is essential for these cells to exert their protective effects, as it enables them to identify pathogens and initiate the necessary immune responses, ultimately contributing to the body’s defense mechanisms [8]. Pattern recognition receptors (PRRs) are specialized proteins which recognize molecular structures associated with pathogens or cellular damage [9]. Such structures include pathogen-associated molecular patterns (PAMPs), which can be found on the surface or within microorganisms like bacteria, viruses, fungi, and parasites. Additionally, damage-associated molecular patterns (DAMPs) act as endogenous signals produced as a result of tissue injury or cellular stress [10].

PRRs are classified into five types based on protein domain homology [11]. Toll-like receptors (TLRs) have the capacity to recognize structurally and biochemically unrelated extracellular ligands, such as lipopolysaccharide (LPS), envelope glycoproteins and heat shock proteins, while C-type lectin receptors (CLRs) recognize mannose, fucose, and glucan carbohydrate structures, which are expressed in many types of microorganisms [12,13]. On the other hand, cytoplasmic PRRs including retinoic-acid-inducible gene-I (RIG-I)-like receptors (RLRs), absent in melanoma-2 (AIM2)-like receptors (ALRs), and NOD-like receptors (NLRs) recognize intracellular ligands. The interaction of PRRs with their ligands is essential to trigger signaling pathways that drive gene expression and possible cytokine and chemokine release [14]. Some PRRs, including NLRP1, NLRP3, and NLRC4, as well as AIM2 and pyrin, have the ability to activate inflammasomes, a group of cytosolic protein complexes that promote inflammation through various mechanisms, including the activation of a type of proteases called caspases [15,16,17].

Inflammasomes generally consist of a sensor protein, usually an NLR or ALR receptor, an adaptor protein and a downstream effector protein [18]. In the NLRP3 inflammasome, NLRP3 is the PRR which serves as the sensor. NLRP3 consists of different domains, each contributing to the activation process. The interaction between PAMPs and DAMPs and the carboxy-terminal leucine-rich-repeat (LRR) domain causes structural changes through a different domain with adenosine triphosphatase (ATPase) activity, called central nucleotide-binding and oligomerization (NACHT) domain. These changes form a complex scaffold for recruiting other inflammasome components [19]. The oligomerized NLRP3′s pyrin domain (PYD) interacts with a different PYD found in an adaptor molecule, the apoptosis-associated speck-like protein containing a caspase recruitment domain (ASC). This interaction causes ASC polymerization and formation of a filamentous structure called ASC speck. The caspase activation and recruitment domain (CARD) of ASC enables the recruitment of pro-caspase-1. Multiple pro-caspase-1 molecules come into proximity, allowing them to auto-proteolytically cleave each other, and turn into activated caspase-1. This effector protein promotes the conversion of pro-inflammatory cytokines, such as pro-interleukin-1β (pro-IL-1β) and pro-interleukin-18 (pro-IL-18), into their bioactive forms, which are then secreted to promote inflammation [20]. Through gasdermin D (GSDMD) cleavage, caspase-1 also triggers pyroptosis, a type of programmed cell death which results in cytomembrane rupture and release of molecules, such as cytokines and DAMPS, that further promote the inflammatory response [21]. The process described above is known as the canonical pathway. However, non-canonical and alternative pathways independent of caspase-1 have also been described [22].

## 3. The Inflammatory Process in Acute Pancreatitis

AP is a common condition of the gastrointestinal tract, responsible for many hospitalizations worldwide. According to the commonly used Atlanta classification, its severity is based on the presence and persistence of organ failure. Acute lung injury, cardiac and kidney dysfunction, and infection of necrotic pancreatic tissue are possible severe results of the inflammatory process in AP. In most cases, a sterile factor initiates pancreatic acinar cell injury. Obstruction of pancreatic fluid outflow due to gallstones and increased ethanol consumption are the most common causes of AP worldwide [23]. After the initial insult causes the release of proteolytic enzymes and pancreatic self-digestion, acinar cells promote the activation of circulating innate immunity cells, including neutrophils and macrophages. These cells are widely accepted as key contributors to the inflammatory cascade in AP, as they aggravate inflammation by producing several cytokines and chemokines [24]. In severe cases, patients might develop a systemic inflammatory response syndrome (SIRS), which can ultimately result in multi-organ dysfunction and death [25].

Different PRRs are involved in the pathogenesis of AP. The interaction of Toll-like receptor 4 (TLR4) with various ligands promotes the infiltration and subsequent damage of pancreatic tissue by innate immune cells, while its deficiency results in decreased AP severity [26]. Zeng et al. found increased expression of Toll-like receptor 9 (TLR9) in mice with cerulein-induced AP [27], while Demirtas et al. observed significantly elevated levels of both TLR9 and NF-κB in blood samples of patients admitted with AP, suggesting its contribution to tissue damage in AP [28]. Activation of the receptor for advanced glycation end products (RAGE) can also affect disease progression in AP. Kang et al. showed that RAGE activation promoted activation of the AIM2 inflammasome and release of cytokines such as IL-1β in macrophages of mice with AP. In the same study, decreased inflammasome activation was observed in RAGE-deficient mice [29].

The interaction of PRRs such as TLRs and RAGE with different ligands activates several signal transduction pathways, including those mediated by nuclear factor kappa-light-chain-enhancer of activated B cells (NF-κB), a central transcription factor. This leads to the expression of genes and production of pro-inflammatory factors which are extensively involved in the inflammation process [30]. The increased presence of cytokines such as IL-1β, IL-6, IL-18, and tumor necrosis factor-a (TNF-a) in AP has been previously demonstrated, and these factors are considered important mediators of tissue damage and necrosis in AP [31,32]. Many of these pathways, as well as the conversion of cytokines such as IL-1β and IL-18 into their mature forms, are associated with activation of the NLRP3 inflammasome.

## 4. The NLRP3 Inflammasome Activation Pathways

### 4.1. The Canonical Pathway

#### 4.1.1. Priming

NLRP3 concentration in cells in a quiescent state appears to be insufficient to cause inflammasome assembly. Thus, a priming signal is required to increase its expression and interaction with different stimuli [33,34]. Inflammasome priming and activating stimuli differ and are specific to disease, tissue, and cell type. Typical priming signals (signal 1) receptors include TLRs and cytokine receptors, such as the tumor necrosis factor receptor (TNFR) and the interleukin 1 receptor type I (IL-1R1) [35]. Myeloid differentiation primary response 88 (MyD88) is a key signaling adaptor which links IL-1R and TLR family members to IL-1R-associated kinase (IRAK) family kinases. Use of the MyD88-dependent pathway results in activation of NF-κB and upregulation of NLRP3 and pro-IL-1β expression [36].

Studies have shown that other post-translational or non-transcriptional mechanisms, including post-translational ubiquitination, phosphorylation, sumoylation, palmitoylation, acetylation, nitrosylation, and ribosylation of the NLRP3 receptor could promote NLRP3 inflammasome priming and activation [37,38,39,40,41]. Fas-associated with death domain protein (FADD), caspase-8 and NOD1/2 are other factors which seem to be involved in NLRP3 inflammasome priming [42,43].

#### 4.1.2. Activation

After priming, a different signal (signal 2) is required to cause NLRP3 inflammasome oligomerization and the subsequent activation of caspase-1. This process depends on many types of stimuli, including electrolyte concentration changes, mitochondrial dysfunction, and lysosomal damage.

Potassium (K^+^) efflux activates the NLRP3 inflammasome with several mechanisms [44]. The formation of pores in cellular membranes from bacterial toxins such as maitotoxin and nigecirin can cause K^+^ efflux. Moreover, adenosine triphosphate (ATP) released from damaged tissue can bind to the P2X7 receptor (P2X7R), an ATP-gated cation-selective channel found in various immune cells, whose activation causes rapid K⁺ efflux. Different types of particulate matter also reduce intracellular K^+^ concentration [45]. K^+^ efflux promotes an open conformation of NLRP3 which leads to inflammasome activation through different processes, interaction of NLRP3 with a protein titled never in mitosis A–related kinase 7 (NEK7) [46]. Sodium (Na^+^) influx can also reduce the K^+^ threshold for NLRP3 activation [47].

Calcium (Ca^2+^) signaling can affect the activation of the NLRP3 inflammasome through various mechanisms, including activation of phosphatidyl inositol/Ca^2+^ pathway by G protein-coupled receptors (GPCR), such as the Ca^2+^-sensing receptor (CaSR), and by facilitating mitochondrial destabilization and increased concentration of reactive oxygen species (ROS) [48,49,50]. Chloride (Cl^−^) channels, including the Cl^−^ intracellular channel (CLIC), also promote NLRP3-Nek7 interaction after their translocation to the plasma membrane due to mitochondrial ROS production [51]. It has been recently demonstrated that inhibition of the with-no-lysine kinase (WNK)-STE20/SPS1-related proline/alanine-rich kinase (SPAK)-Na^+^:K^+^:2Cl^−^ cotransporter 1 (NKCC1) pathway, which senses and maintains both Cl^−^ and K^+^ intracellular concentrations, aggravates NLRP3 inflammasome activation. This further showcases the importance of electrolyte disturbances as NLRP3 activation signals [52].

Mitochondrial dysfunction and inhibition of protective cellular processes such as mitophagy, during which dysfunctional mitochondria are removed, cause excessive ROS production [53]. Accumulation of mitochondrial ROS, mitochondrial deoxyribonucleic acid (DNA), and other mitochondria-associated proteins are capable of causing the assembly of the NLRP3 inflammasome. Imbalance of mitochondrial dynamics, such as fission and fusion, can also threaten mitochondrial homeostasis [54].

Lysosomes are involved in the degradation of many endogenous and exogenous substances. Several molecules, including silica, Ca^2+^ and cholesterol crystals, once phagocytosed, can cause a decrease in lysosomal pH and lysosomal membrane destabilization, leading to the release of contents such as cathepsin B, an enzyme involved in the regulation of IL-1β release [55]. The accumulation of lysosomal proteases in the cytosol leads to K^+^ efflux through increased permeability of the cytoplasmic membrane [56]. These events promote NLRP3 inflammasome activation.

#### 4.1.3. Pyroptosis

Pyroptosis is a type of cell death activated by different types of caspases through cleavage of a 53 kDa substrate called gasdermin D (GSDMD). GSDMD consists of an *N*-terminal domain with pore-forming ability, a *C*-terminal domain with regulatory function, and a central linker region. Before cleavage, activation of the *N*-terminal domain is inhibited by the *C*-terminal domain [57]. Once cleaved, the *N*-terminal fragment of GSDMD translocates to the cytoplasmic membrane to form pores. These pores cause cell swelling and eventual disruption of the integrity of the cell membrane, leading to cell lysis. This allows the release of intracellular molecules, including pro-inflammatory cytokines, ATP and high mobility group box 1 (HMGB1), a nuclear protein which acts as a DAMP, regulating inflammation and immune responses when increased in the cytosol [58,59].

### 4.2. Non-Canonical and Alternative Pathways

Non-canonical activation of the NLRP3 inflammasome is a different pathway, mainly initiated by Gram-negative bacteria. LPS, a substance found in the outer membrane of Gram-negative bacteria, is recognized by TLR4. Following association with LPS, TLR4 signaling through the MyD88 and Toll/IL-1 receptor homology-domain-containing adapter-inducing interferon-β (TRIF) pathways initiates translocation of NF-κB into the nucleus. This promotes the transcription of genes of pro-inflammatory cytokines as well as interferon regulatory factor (IRF)-3 and IRF7 genes. The IRF3–IRF7 complex activates the Janus kinase/signal transducers and activators of transcription (JAK/STAT) pathway, which leads to production of caspase-11 in mice (or caspase-4/5 in humans) [60,61]. A direct interaction between LPS and caspase-11 triggers caspase-11 oligomerization and induces its catalytic activity [62]. Since caspase-11 cannot process pro-IL-1β and pro-IL-18 but can induce cleavage of GSDMD, non-canonical inflammasomes mainly induce pyroptosis. Cross-talk between non-canonical and canonical inflammasome activation pathways, such as the activation of pannexin-1 by caspase-11 and subsequent release of ATP and activation of P2X7R to induce K^+^ efflux and thus canonical NLRP3 assembly, has also been suggested [63].

The alternative inflammasome pathway refers to a distinct activation mechanism of the NLRP3 inflammasome (Figure 1). It involves the direct activation of the NLRP3 inflammasome without the need for classical triggers, such as K^+^ efflux or mitochondrial damage, which typically prime and activate NLRP3. Gaidt et al. found that NLRP3 activation was mediated by TLR4-TRIF-receptor-interacting serine/threonine-protein kinase 1 (RIPK1)-FADD-caspase-8 pathway; however, pyroptosis was not induced. Since caspase-8 did not cleave NLRP3, the authors speculated that the cleavage of an unknown intermediate protein by caspase-8 is necessary for the alternative activation of the inflammasome [64]. It has also been suggested that TLR ligands alone could stimulate IL-1β production by innate immune cells. Activation of TLR receptors in an in vitro model of primary human monocytes induced caspase-1 activation and IL-1β release even in the absence of a secondary signal. This indicates that NLRP3 activation does not necessarily require TRIF and RIPK1 activity [65].

Figure 1 illustrates the pathways leading to nucleotide oligomerization domain (NOD)-like receptor family pyrin domain-containing 3 (NLRP3) inflammasome activation, including the canonical, non-canonical, and alternative pathways. Signal 1 (priming) is initiated by receptors such as tumor necrosis factor receptor (TNFR), interleukin 1 receptor (IL-1R), and TLRs, activating NF-κB, which induces the transcription of NLRP3 and pro-inflammatory cytokines (pro-IL-1β and pro-IL-18). In the canonical pathway (Signal 2), danger signals like K⁺ efflux, mitochondrial ROS (mtROS), and lysosomal damage lead to NLRP3 oligomerization, recruitment of adaptor-apoptosis-associated speck-like protein (ASC), never in mitosis A–related kinase 7 (NEK7), and pro-caspase-1, and the formation of the NLRP3 inflammasome. Caspase-1 activation cleaves pro-IL-1β and pro-IL-18 into their mature forms, triggering the release of cytokines and inducing pyroptosis through cleavage of GSDMD (gasdermin D). The non-canonical pathway involves the activation of caspase-11 (mouse), or caspases-4 and -5 (human), in response to intracellular lipopolysaccharide (LPS), which leads to gasdermin D-mediated pyroptosis. The alternative pathway involves Toll-like receptor 4 (TLR4) signaling through Toll/IL-1 receptor homology-domain-containing adapter-inducing interferon-β (TRIF) and the activation of caspase-8, contributing to inflammasome activation without direct involvement of K⁺ efflux. Created with BioRender.com.

## 5. Contribution of the NLRP3 Inflammasome in the Pathogenesis of Acute Pancreatitis

### 5.1. NLRP3 Inflammasome Is Activated During Acute Pancreatitis

As mentioned above, many cytokines, including IL-1 and IL-18, are involved in the pathogenesis of AP. The precursor forms of these two molecules are cleaved by caspase-1, the effector protein of the canonical NLRP3 inflammasome. This makes NLRP3 an important mediator of the inflammatory process in AP. Fu et al. demonstrated that NLRP3 deficiency reduced neutrophil infiltration and the severity of tissue injury to the pancreas and lungs in a cerulein plus LPS-induced AP model [66]. Hoque et al. observed reduced disease severity in NLRP3-deficient mice with cerulein-induced AP [67]. Increased concentrations of NLRP3 inflammasome components have also been found both in the early and advanced stages of AP. Chueca et al. found increased levels of caspase-1, IL-1β and IL-18 in PBMCs isolated from blood samples of AP patients during the first day after hospital admission [68]. Increased expression of the NLRP3 gene, as well as other components of the inflammasome was found in macrophages of mice with severe AP. In the same study, NLRP3-deficient mice had reduced tissue damage and fibrogenesis due to a decreased inflammatory response by both innate and adaptive immune cells many days after disease initiation [69]. These studies highlight the crucial contribution of the NLRP3 inflammasome in the inflammatory cascade in AP (Table 1).

### 5.2. Factors That Promote Activation of Nlrp3 Inflammasome in Acute Pancreatitis

#### 5.2.1. DAMP Signals

Many molecules act as DAMPs and promote the inflammatory process in AP. Among these, extracellular HMGB1 plays a pivotal role in the exacerbation of inflammation during AP. HMGB1 interacts with RAGE and TLRs, including TLR4, and promotes inflammation through various mechanisms [77]. Li et al. observed increased HMGB1 levels in mice with AP, as well as dose-dependent pancreatic injury after external HMGB1 administration in mice, mainly attributed to interaction with TLR4 and elevated NF-κB activation. These findings were less evident in TLR4-deficient mice, further supporting the contribution of HMGB1 to the inflammatory process in AP [70]. Wu et al. examined the effect of HMGB1 administration in an in vitro model of human-derived neutrophils and macrophages, as well as in an AP mouse model. HMGB1 administration caused increased formation of neutrophil extracellular trap (NET) and elevated expression of NLRP3 and IL-1 genes, leading to increased activation of the canonical NLRP3 pathway, thus aggravating cell damage in AP [71].

Heat Shock Protein 70 (Hsp70) is a cytoprotective molecular chaperone found in all organisms, which is involved in protein folding, repair and protection against stress-induced damage. A protective role and increased expression of Hsp70 in AP have also been suggested [78]. Chen et al. recently demonstrated that increased expression of Hsp70 in mice with sodium taurocholate-induced AP resulted in reduced pancreatic inflammation and tissue damage [79]. However, extracellular Hsp70 (eHsp70) can act as a DAMP and promote inflammation in many conditions [80]. Song et al. observed increased pancreatic NF-κB activity and aggravation of AP severity and SIRS after administration of eHsp70 in mice with cerulein-induced AP. The lack of a similar effect in TLR-4-deficient mice suggested that eHsp70 might activate the MyD88/IRAK/NF-κB signal transduction pathway by interacting with TLR4, leading to increased cytokine production, such as IL-1α [72]. Despite the lack of solid evidence directly linking eHsp70 to NLRP3 inflammasome activation, the increased presence of eHsp70 in patients with AP and the use of similar signal transduction pathways make the interaction between eHsp70 and NLRP3 a reasonable hypothesis.

ATP is a source of high energy for many cell types. However, its release in the extracellular space after cellular damage promotes the inflammatory process. Its interaction with P2X7R, a potent activator of the NLRP3 inflammasome, induces mitochondrial damage and causes production of IL-1β and IL-18 [81]. Dixit et al. demonstrated that blocking the effect of extracellular ATP (eATP) reduced levels of inflammatory cytokines and pancreatic and lung injury in mice with AP. The results support the role of eATP and purinergic signaling in the inflammatory process during AP [73].

#### 5.2.2. Bacterial Translocation

Bacterial infiltration of the inflamed pancreas is one of the most severe complications of AP. The intestinal mucosa acts as a barrier and maintains the stability of the intestinal environment. A persistent inflammatory response, such as the one seen in severe AP, damages the intestinal mucosa, leading to alterations of the composition of the intestinal microbiome [82]. Activation of pathways important for NLRP3 assembly, including binding to TLR4 and use of the MyD88/IRAK/NF-κB signal transduction pathway and subsequent cytokine release, appear to damage the intestinal barrier and aggravate inflammation in AP [83]. Studies have also shown that inter-intestinal probiotics can reduce the severity of AP by inhibiting the activation of the NLRP3 inflammasome in the gut [74]. Promotion of NOD1 signaling from PAMPs deriving from circulating microorganisms might also contribute to the development of AP. Co-administration of low-dose cerulein with a NOD1 ligand caused a severe form of AP by promoting production of monocyte chemotactic protein-1 (MCP-1) through NF-κB and signal transducers and activators of transcription 3 (STAT3) signaling in an animal model [84]. Binding of diaminopimelic acid (DAP), a molecule found in bacterial cell walls, to NOD1 and subsequent activation of the NOD1/ receptor-interacting protein 2 (RIP2)/NF-κB pathway also appear to aggravate inflammation in AP [75]. These findings further support the notion that bacterial translocation may contribute to NLRP3 activation.

#### 5.2.3. Cathepsins

Cathepsins are a family of proteolytic enzymes, primarily found in lysosomes. They play various roles in cellular processes, including protein degradation. Cathepsins, especially cathepsin B (CTSB), have been highlighted as key components of NLRP3 activation [85]. In AP, stress or injury to pancreatic cells leads to the release of lysosomal contents, including CTSB (Figure 2). CTSB activates the NLRP3 inflammasome pathway and induces caspase-1 activation, thus leading to secretion of IL-1β and IL-18. Pyroptosis is another mechanism by which cathepsins cause pancreatic cell damage and aggravation of the inflammatory process in AP [76].

Figure 2 depicts the mechanisms involved in nucleotide oligomerization domain (NOD)-like receptor family pyrin domain-containing 3 (NLRP3) inflammasome activation during AP. Stress or injury to pancreatic cells, caused by various factors such as pathogen-associated molecular patterns (PAMPs) or damage-associated molecular patterns (DAMPs), triggers Toll-like receptor (TLR) activation. TLR signaling through nuclear factor kappa-light-chain-enhancer of activated B cells (NF-κB) leads to the transcription of NLRP3, pro-interleukin-18 (pro-IL-18), and pro-IL-1β. Lysosomal damage releases cathepsin B, while mitochondrial damage results in the production of mitochondrial reactive oxygen species (mtROS) and the release of oxidized mitochondrial DNA (ox-mtDNA), contributing to NLRP3 inflammasome activation. Additionally, adenosine triphosphate (ATP) release activates the P2X7R, promoting potassium efflux, further amplifying inflammasome assembly. NLRP3 associates with ASC and pro-caspase-1, leading to the activation of caspase-1, which processes pro-IL-18 and pro-IL-1β into their mature forms. Caspase-1 also cleaves gasdermin D (GSDMD), generating N-GSDMD, which forms pores in the cell membrane, allowing the release of pro-inflammatory cytokines (IL-18, IL-1β), cell swelling, and pyroptotic cell death. In the non-canonical pathway, caspase-11 (in mice) and caspases-4 and -5 (in humans) respond to intracellular lipopolysaccharide (LPS), further promoting pyroptosis. Figure 2 also highlights bacterial translocation and altered microbiota as contributing factors in the progression of acute pancreatitis. Interactions of PAMPs with different receptors, including binding of diaminopimelic acid (DAP) to the nucleotide-binding oligomerization domain 1 (NOD1), lead to acinar cell injury, inflammation, cytokine release, and eventual cell rupture. Created with BioRender.com

### 5.3. NLRP3 Activation and the Resulting Effect of Cytokines in AP

#### 5.3.1. IL-1β

The IL-1 family of cytokines consists of 11 members, all of which are involved in regulating inflammatory responses to microorganisms and sterile insults. IL-1 promotes sterile inflammatory processes by various mechanisms, including direct tissue destruction and production of cytokines such as IL-6 and TNFα, after binding to its receptor [86]. In AP, IL-1β is secreted by different types of leucocytes after activation of the NLRP3 inflammasome and is one of the prime mediators of the inflammatory process. Results from a ceruline-induced AP model indicated that IL-1β could reduce the viability of acinar cells by inducing the activation of trypsinogen to trypsin and impairing autophagy [87]. Studies showcasing the protective effect of IL-1 receptor antagonists in AP further demonstrate the participation of IL-1 in the inflammatory process in AP [88].

#### 5.3.2. IL-18

IL-18, best known for its ability to induce IFN-γ, is believed to play an important role in various inflammatory conditions by inducing both Th1 and Th2 immune responses, as well as activating innate immune cells. IL-18 has an important role in pancreatic diseases, including AP [31]. Its elevated presence in human patients and animal models of AP showcases the involvement of IL-18 in the pathogenesis of AP [89,90]. Co-administration of IL-18 with IL-12 caused severe pancreatic inflammation with adipose tissue necrosis and saponification in an animal model, highlighting the importance of the interaction between IL-18 and other pro-inflammatory substances [91]. After being cleaved by the NLRP3 inflammasome, IL-18 interacts with the IL-18 receptor, triggering NF-κB activation through the MyD88-dependent pathway [92]. A clear correlation of NLRP3 inflammasome activation and AP severity has been demonstrated by Sendler et al. NLRP3- or IL-18-deficient mice with acute AP had reduced activation of T cells and no increase in Th2 cell-mediated responses compared to controls. The results showed that SIRS and compensatory anti-inflammatory response syndrome (CARS) occurred simultaneously rather than sequentially [69].

#### 5.3.3. IL-33

Many other cytokines play important roles in the pathogenesis of AP; however, the involvement of the NLRP3 inflammasome in their processing and release is not yet clearly demonstrated. IL-33 is a critical member of the IL-1 family, particularly found in the nuclei of epithelial and endothelial cells. It signals through its ligand-binding primary receptor ST2 and IL-1 receptor accessory protein (IL-1RAcP). Tissue damage can cause the release of IL-33 into the extracellular environment, where it functions as an endogenous danger signal [93]. In a taurocholate-induced AP model, IL-33 production was associated with high disease severity, while its levels increased after injection of TNF-a. The authors suggested that the increase in IL-33 levels due to TNF-a and subsequent IL-6 secretion through IL-33/sST2 signaling might contribute to the inflammatory process in AP [94]. Similarly to IL-18, IL-33 signaling can activate the MyD88/IRAK/NF-κB axis and promote NLRP3 production [95]. However, caspase-1 appears to inactivate IL-33 [96], while a protective role for ST2 during AP has been suggested [97]. These findings make IL-33 association with NLRP3 and its possible role in AP a topic for further investigation.

## 6. The NLRP3 Inflammasome as a Prognostic Factor in Acute Pancreatitis

NLRP3 is a promising biomarker for pancreatitis patients. Despite limited research involving human patients, data suggest that NLRP3 itself and inflammasome-related products of this pathophysiological procedure can serve as useful markers for such patients. Chueca et al. found that NLRP3 activation leads to a significant increase in IL-1β and IL-8, key products of the NLRP3 inflammasome pathway, when compared to controls. In the same study, CRPmax in 48 h was positively associated with disease severity. While NLRP3 showed a positive correlation with CRPmax, this relationship did not reach statistical significance. NLRP3 was also found increased in patients with severe AP compared to mild AP and healthy controls but this increase was also not statistically significant [68]. IL-1β was demonstrated as a key product in the NLRP3 inflammasome pathway in patients with acute lung injury (ALI) related to severe AP. This result suggests IL-1β as a potential biomarker for these patients [98]. In a mouse model of AP, the increase in NLRP3 was statistically significant when severe AP was induced [99]. Another mouse model also links the increase in NLRP3 to the severity of AP and the related ALI [66]. Lung myeloperoxidase (MPO) is a marker related to the aforementioned ALI. An imbalance in interleukins secreted by PBMCs, favoring IL-4 and IL-10 over IL-17 and IFN-γ, has also been demonstrated in a mouse model [69]. The increased expression of NLRP3 in patients with severe AP also causes elevated kidney MPO levels and triggers a cytokine cascade, contributing to renal damage and acute kidney injury (AKI) in severe AP models [100].

NLRP3 has also been tested as a marker for pancreatic cancer prognosis. It has been demonstrated, that overproduction of NLRP3 inflammasome and its products can predict reduced survival and advanced stage of the disease [101,102]. Consequently, it would be promising to also examine those products as severity markers for AP. NLRP3 inflammasome additionally activates neutrophil extracellular trap (NET) formation leading to IL-1β overproduction. This is a key pathophysiologic mechanism of SIRS, a very frequent complication in AP patients. Thus, NETs, including their sub-particles and final products, could act as a predictive marker for SIRS in severe forms of AP [103,104].

## 7. The NLRP3 Inflammasome as a Therapeutic Target in Acute Pancreatitis

The strong inflammatory potential and its important role in causing cell damage have made the NLRP3 inflammasome an attractive target for the treatment of various conditions. To date, however, clinical application of this knowledge remains limited, as most studies regarding NLRP3 inhibition have been conducted in animal models (Table 2). A recent systematic review by Zhang et al. showcased promising roles for NLRP3 inhibitors in the treatment of NLRP3-inflammasome-associated inflammatory diseases. However, the authors noted that further research is required to confirm the effectiveness of these agents, before they can be implemented in clinical practice [105]. Another systematic review by Gao et al., which included twenty-eight animal studies involving 556 animals with AP, showed that NLRP3 inflammasome inhibition has protective effects on AP by reducing local pancreatic injury, systemic inflammatory responses and organ dysfunction. They concluded that inhibition of NLRP3 inflammasome is a promising therapeutic strategy for the treatment of AP [106].

To date, clinical treatments for diseases involving the NLRP3 inflammasome predominately target IL-1β using antibodies or recombinant IL-1β receptor antagonists, such as Anakinra, Rilonacept and Canakinumab [124]. The effectiveness of Anakinra in treating AP was shown in an animal model, in which Anakinra injection significantly decreased pancreatic tissue injury and apoptosis [107]. However, inhibition of IL-1 could cause severe immunosuppression, due to its involvement in other inflammatory pathways besides NLRP3 inflammasome. Thus, pharmacological agents specifically targeting the NLRP3 inflammasome may be a safer and more effective choice for NLRP3-related diseases [34].

MCC950 is an effective and selective inhibitor of NLRP3 inflammasome, which reduces its formation mainly by eliminating ASC oligomerization [125]. It is a diarylsulphonylurea-containing compound which does not seem to affect the activation of other inflammasomes [126]. Sendler et al. examined the use of MCC950 in experimental AP, with results showing reduction in IL-18 levels and disease severity [69]. Administration of MCC950 reduced damage of both pancreatic cells and intestinal barrier by inhibiting the NLRP3 inflammasome in mice with AP [108]. Glyburide, another sulphonyluria drug, reduced pancreatic damage and cell death in genetically obese mice with AP by blocking K^+^ channels in the cell membrane and thus preventing NLRP3 inflammasome activation [109]. IFN-39, a direct NLRP3 inhibitor, reduced pancreatic damage and the associated acute lung injury in a cerulein plus LPS-AP animal model [66]. Further studies are required to further examine the specificity and safety profile of these agents in AP.

Some natural plant extracts have also been used in the treatment of experimental AP. A systematic review by Tang et al. included experimental studies of 30 phytochemicals with potential therapeutic effects in AP. They concluded that plant products with anti-inflammatory and antioxidant abilities may be efficient candidate drugs for AP treatment [127]. Some of these phytochemicals appear to apply their anti-inflammatory actions through NLRP3 inflammasome inhibition. Emodin, a natural antraquinone product, reduced pancreatic injury and systemic inflammation by inhibiting the P2X7/NLRP3 signaling pathway in mice with taurocholate-induced AP [110]. In a similar animal model, emodin prevented severe AP-associated lung injury by inhibiting NF-Κb and increasing the nuclear translocation of nuclear factor erythrocyte-2 associated factor 2 (Nrf2), thus promoting Nrf2/heme oxygenase-1 (HO-1) signaling [111]. Rutin, a flavonoid derivative, reduced the severity of the inflammatory process in an animal AP model by limiting ASC expression in the pancreas and thus preventing activation of caspase-1 [112]. Kanak et al. observed reduced translocation of NF-κB and limited endoplasmic reticulum stress after administration of withaferin A, a small steroidal lactone alkaloid, to mice with AP [113].

Cyclooxygenase-2(COX-2) inhibitors reduce the production of inflammatory prostaglandins, thus alleviating pain and inflammation. Their preventive effect against pancreatitis has been demonstrated in clinical practice [128]. Indomethacin reduced the expression of IL-1β and subsequent pancreatic damage by inhibiting the activation of NLRP3 inflammasome in a cerulein plus LPS-induced AP animal model [114]. Iguratimod (T-614) was confirmed to inhibit the NF-κB signaling pathway and limit NLRP3 inflammasome activation in an animal study [115]. These studies suggest a possible use for COX-2 inhibitors in patients with AP.

N-(3′,4′-dimethoxycinnamonyl) anthranilic acid (3,4-DAA), a tryptophan metabolite, reduced cell damage by inhibiting NF-κΒ activation in a cerulein plus LPS-induced AP animal model [116]. Cordycepin, a plant derivative with anti-inflammatory capabilities, also appeared to suppress NF-κΒ and subsequent NLRP3 inflammasome assembly by causing phosphorylation of activated protein kinase (AMPK) in mice with AP [117]. Ren et al. found a reduction in ROS production, NF-κΒ and NLRP3 activation after administration of hydrogen-rich saline in an animal model, which resulted in reduced pancreatic injury [118].

INT-777, a molecule which acts as an agonist of Takeda G-protein coupled receptor clone 5 (TGR5), a bile acid receptor, has been shown to inhibit the NLRP3 inflammasome in many inflammatory diseases [129]. In a cerulein-induced AP model, decreased ROS production, subsequent reduction in NLRP3 activation and limited pancreatic damage were observed after administration of INT-777, making a protective use for this agent in patients with AP a possibility [119]. Similar protective effects for both pancreatic and lung tissue were observed after administration of apocynin, an inhibitor of nicotinamide adenine dinucleotide phosphate (NADPH) synthase (NOX), to rats with AP. These effects were mainly attributed to decreased ROS production and NF-κB/ NLRP3 signaling [120]. Administration of Rg100204, an aquaporin-9 (AQP9) inhibitor, reduced pancreatic and lung damage in animals with AP by interacting with the NF-κB and Nrf2 pathways and thus suppressing NLRP3 activation [99].

β-hydroxybutyric acid, a ketone body produced by oxidation of fatty acids in the liver, can decrease cellular stress by reducing ROS production. Administration of β-hydroxybutyric acid in mice with AP resulted in low expression of many NLRP3 inflammasome components, including NLRP3 and ASC, and led to low cytokine production and reduced tissue damage [121]. Butyrate, a short chain fatty acid with anti-inflammatory effects, was examined by Pan et al., who showcased that prophylactic butyrate administration reduced NLRP3 activation by limiting phosphorylation of transcription factors, including NF-Κβ, activator protein 1 (AP1), signal transducers and activators of transcription 1 (STAT1) and STAT3 in mice with cerulein-induced AP, after binding to histone deacetylase 1 (HDAC1) and GPR109A receptor in the pancreas and colon, respectively [122]. These studies suggest a possible beneficial use of these agents in AP treatment.

Many of the agents mentioned above show promise in AP treatment through inhibition of NF-κΒ signaling. However, due to its role as a mediator of many important immune responses, total inhibition of NF-κB signaling can cause severe side effects, including immunosuppression, hepatotoxicity and development of malignant tumors [130]. This makes the development of tissue-specific NF-κB inhibitors a possible target for future clinical trials.

As discussed above, bacterial translocation often occurs in AP after disruption of the integrity of the intestinal barrier, contributing to disease severity by activating the NLRP3 inflammasome. Jia et al. examined the effect of antibiotic combination therapy consisting of vancomycin, neomycin, and polymyxin b in a cerulein-induced AP animal model. They observed reduced pancreatic inflammation after antibiotic administration, which was attributed to inhibition of the TLR4/NLRP3 pathway [123]. However, routine use of antibiotics in the treatment of AP is not currently recommended, and complications such as the development of antibiotic resistance and fungal infections are a major concern [131]. Future studies are needed to evaluate which patients with AP will mostly benefit from antibiotic administration and the resulting NLRP3 inhibition.

## 8. Conclusions

AP continues to be a therapeutic challenge for modern clinicians. NLRP3 inflammasome has proven to be an important mediator of the inflammatory process in the pathogenesis of this disease, as its activation causes production of pro-inflammatory cytokines and pyroptosis, leading to severe pancreatic damage. Many different molecules and disruption of cell homeostasis can cause the assembly of the NLRP3 inflammasome through different pathways. The variety of molecules and pathways involved in the activation of NLRP3 inflammasome and promotion of the inflammatory process have made it a reasonable therapeutic target in AP. Studies on many animal models support its use as a prognostic marker in patients with AP, while a variety of NLRP3 inhibitors are showing promise in AP treatment and prevention of disease progression. However, the prognostic and therapeutic effects of the NLRP3 inflammasome and its inhibitors have not been demonstrated in the clinical context. Many agents who inhibit the NLRP3 pathway do not act specifically in the context of AP and could cause severe side effects. Future studies and clinical trials are needed to investigate the possible effects of NLRP3 inhibitors in patients with AP, the optimal dose and regimen of each agent, as well as their specificity and safety profile. The prognostic value of NLRP3 in the context of AP is a subject which also requires further clinical investigation. Such trials could have an important impact on AP treatment and help improve patient care.

## Figures and Tables

**Figure 1 biology-13-00945-f001:**
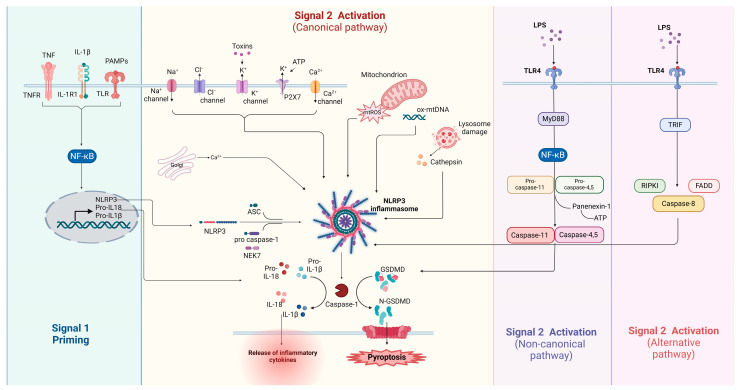
NLRP3 inflammasome activation pathways.

**Figure 2 biology-13-00945-f002:**
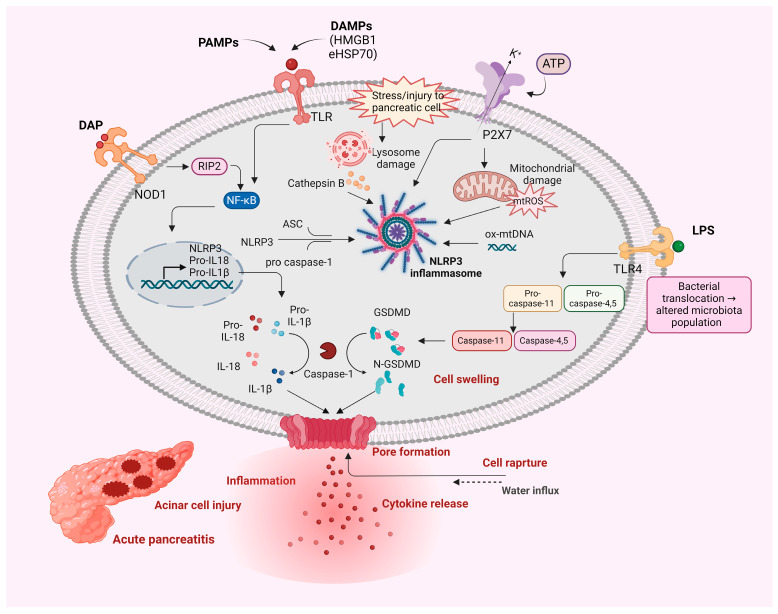
NLRP3 inflammasome activation in acute pancreatitis.

**Table 1 biology-13-00945-t001:** Studies that support the role of NLRP3 as a therapeutic target in acute pancreatitis.

Title	Author	Year	NLRP3 Inflammasome Association with AP
NLRP3 deficiency alleviates severe acute pancreatitis and pancreatitis-associated lung injury in a mouse model [66].	Fu, Q.	2018	Reduced pancreatic and lung damage in NLRP3-deficient mice
TLR9 and the NLRP3 inflammasome link acinar cell death with inflammation in acute pancreatitis [67].	Hoque, R.	2008	Reduced pancreatic damage in NLRP3-deficient mice
The expression and activation of the AIM2 inflammasome correlates with inflammation and disease severity in patients with acute pancreatitis [68].	Chueca, A.F.	2017	Increased caspase-1, IL-1β and IL-18 in PBMCs isolated from blood samples of AP patients
NLRP3 inflammasome regulates development of systemic inflammatory response and compensatory anti-inflammatory response syndromes in mice with acute pancreatitis [69].	Sendler, M.	2020	Increased expression of NLRP3 by macrophages in mice with AP, reduced tissue damage in NLRP3-deficient mice
TLR4-mediated NF-κB signaling pathway mediates HMGB1-induced pancreatic injury in mice with severe acute pancreatitis [70].	Li, G.	2015	Activation of TLR4/NF-κB signaling pathway by HMGB1
High-mobility group box protein-1 induces acute pancreatitis through activation of neutrophil extracellular trap and subsequent production of IL-1β [71].	Wu, X.	2021	Pancreatic cell damage after activation of NLRP3 by HMGB1 and increased production of IL-1β
Extracellular heat-shock protein 70 aggravates cerulein-induced pancreatitis through toll-like receptor-4 in mice [72].	Song, J.M.	2008	Pancreatic injury due to possible activation of TLR4/NF-κB signaling pathway by eHsp70
Extracellular release of ATP promotes systemic inflammation during acute pancreatitis [73].	Dixit, A.	2019	Activation of NF-κB and elevated expression of NLRP3, caspase-1 and IL-1β at increased concentration of eATP
Clostridium butyricum Strains Suppress Experimental Acute Pancreatitis by Maintaining Intestinal Homeostasis [74].	Pan, L.L.	2019	Reduced pancreatic damage due to suppression of TLR4 signaling and NLRP3 inflammasome activation by probiotics
Gut Microbiota-Derived Diaminopimelic Acid Promotes the NOD1/RIP2 Signaling Pathway and Plays a Key Role in the Progression of Severe Acute Pancreatitis [75].	Jiao, J.	2022	Activation of NF-κB after DAP recognition by NOD1
Cathepsin B aggravates acute pancreatitis by activating the NLRP3 inflammasome and promoting the caspase-1-induced pyroptosis [76].	Wang, J.	2021	Increased pancreatic injury due to pyroptosis after NLRP3 activation by CTSB

Abbreviations: NLRP3, NLR pyrin domain-containing protein 3; AP, acute pancreatitis; TLR9, Toll-like receptor 9; IL-1β, interleukin 1β; IL-18, interleukin 18; NF-κB, nuclear factor kappa B; HMGB1, high mobility group box 1; eHsp70, extracellular Heat Shock Protein 70; eATP, extracellular adenosine triphosphate; DAP, diaminopimelic acid; NOD1, nucleotide-binding oligomerization domain 1; CTSB, cathepsin B.

**Table 2 biology-13-00945-t002:** Studies examining the effect of NLRP3 inhibitors in acute pancreatitis.

Title	Author	Year	NLRP3 Inhibitor	Target
Effectiveness of interleukin-1 receptor antagonist (Anakinra) on cerulein-induced experimental acute pancreatitis in rats [107].	Kaplan, M.	2014	Anakinra	IL-1
NLRP3 inflammasome regulates development of systemic inflammatory response and compensatory anti-inflammatory response syndromes in mice with acute pancreatitis [69].	Sendler, M.	2020	MCC950	ASC oligomerization
NLRP3 inflammasome inhibitor MCC950 can reduce the damage of pancreatic and intestinal barrier function in mice with acute pancreatitis [108].	Shen, Y.	2022	MCC950	ASC oligomerization
Inhibition of the nucleotide-binding domain, leucine-richcontaining family, pyrin-domain containing 3 inflammasome reduces the severity of experimentally induced acute pancreatitis in obese mice [109].	York, J.M.	2014	Glyburide	ATP-sensitive K^+^ channels
NLRP3 Deficiency Alleviates Severe Acute Pancreatitis and Pancreatitis-Associated Lung Injury in a Mouse Model [66].	Fu, Q.	2018	INF-39	NLRP3
Emodin attenuated severe acute pancreatitis via the P2X ligand-gated ion channel7/NOD-like receptor protein 3 signaling pathway [110].	Zhang, Q.	2019	Emodin	P2X7/NLRP3
Emodin Protects Against Acute Pancreatitis-Associated Lung Injury by Inhibiting NLPR3 Inflammasome Activation via Nrf2/HO-1 Signaling [111].	Gao, Z.	2020	Emodin	NF-κB/Nrf2
Rutin modulates ASC expression in NLRP3 inflammasome: a study in alcohol and cerulein-induced rat model of pancreatitis [112].	Aruna, R.	2014	Rutin	ASC/caspase-1
A small molecule inhibitor of NF-κB blocks ER stress and the NLRP3 inflammasome and prevents progression of pancreatitis [113].	Kanak, M.A.	2017	Withaferin A	NF-κB/NLRP3
Indomethacin inhabits the NLRP3 inflammasome pathway and protects severe acute pancreatitis in mice [114].	Lu, G.	2017	Indomethacin	NLRP3/ASC/IL1-β
Iguratimod (T-614) attenuates severe acute pancreatitis by inhibiting the NLRP3 inflammasome and NF-κB pathway [115].	Hou, C.	2019	Iguratimod	NF-κB/NLRP3
N-(3′,4′-dimethoxycinnamonyl) anthranilic acid alleviates severe acute pancreatitis by inhibiting intestinal barrier dysfunction and NF-κB activation [116].	Zhao, Z.	2021	N-(3′,4′-dimethoxycinnamonyl) anthranilic acid	NF-κB/NLRP3
Cordycepin protects against acute pancreatitis by modulating NF-κB and NLRP3 inflammasome activation via AMPK [117].	Yang, J.	2020	Cordycepin	AMPK/NF-κB/NLRP3
Hydrogen-rich saline inhibits NLRP3 inflammasome activation and attenuates experimental acute pancreatitis in mice [118].	Ren, J.D.	2014	Hydrogen-rich saline	ROS/NF-κB
INT-777, a bile acid receptor agonist, extenuates pancreatic acinar cells necrosis in a mouse model of acute pancreatitis [119].	Li, B.	2018	INT-777	ROS/NLRP3
Apocynin alleviates lung injury by suppressing NLRP3 inflammasome activation and NF-κB signaling in acute pancreatitis [120].	Jin, H.Z.	2019	Apocynin	ROS/NF-κB/NLRP3
Inhibition of aquaporin-9 ameliorates severe acute pancreatitis and associated lung injury by NLRP3 and Nrf2/HO-1 pathways [99].	Chen, J.	2024	Rg100204	AQP9/NF-κB/Nrf2
β-Hydroxybutyrate, one of the three main ketone bodies, ameliorates acute pancreatitis in rats by suppressing the NLRP3 inflammasome pathway [121].	Şahin, E	2021	β-Hydroxybutyrate	NF-κB/NLRP3
Butyrate ameliorates caerulein induced acute pancreatitis and associated intestinal injury by tissue-specific mechanisms [122].	Pan, X.	2019	Butyrate	NF-κB/STAT1/STAT3/AP1
Combinatory antibiotic treatment protects against experimental acute pancreatitis by suppressing gut bacterial translocation to pancreas and inhibiting NLRP3 inflammasome pathway [123].	Jia, L.	2020	Vancomycin, neomycin, and polymyxin b	TLR4/NLRP3

Abbreviations: NLRP3, NLR pyrin domain-containing protein 3; IL-1, interleukin 1; ASC, caspase recruitment domain; ATP, adenosine triphosphate; NF-κB, nuclear factor kappa B; Nrf2, nuclear factor erythrocyte-2 associated factor-2; AMPK, activated protein kinase; ROS, reactive oxygen species; AQP9, aquaporin-9; AP1, activator protein 1; STAT1, signal transducers and activators of transcription 1; STAT3, signal transducers and activators of transcription 3.

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
