# Peer review of "Understanding the Role of NLRP3 Inflammasome in Acute Pancreatitis"

_biology, 2024, doi:10.3390/biology13110945_

Round 1

Reviewer 1 Report

Comments and Suggestions for Authors

In this review the authors discuss the role of the NLRP3 inflammasome as a mediator of Acute Pancreatitis (AP). They also evaluate the potential of a few NLRP3 inflammasome inhibitors as a treatment option for AP and the possibility of the NLRP3 inflammasome as a biomarker for the prediction and prognosis of AP. This review is very insightful, however is extremely similar to the previously published review by Ferrero-Andres et al, 2020. A broader discussion on NLRP3 inflammasome inhibition in AP is needed to strengthen this study and build on the work of Ferrero-Andres et al; 2020. 

I recommend this review article for publication after major revisions which are as follow:

Point 1: Please add a little bit of information on how AP is currently treated and why these treatments fail.

Point 2: sections 2 and 3.1 are very similar and lots of overlapping material, this could be condensed into 1 section.

Point 3: In line 146 authors state “The expression level of pro-IL-18 does not seem to be influenced by priming” however in line 244 authors have written that NF-kB promotes transcription of IL-18”, please revisit this.

Point 4:  line 172 - define calcium (Ca2+)  in first instance 

              line 177 - define chloride (Cl-) here 

Point 5: line 211 - 215 - this is repetitive of the previous paragraph and can be removed or condensed.

Point 6: section 3.2.1 and section 3.2.2 - I think it is important to note here that inflammasome priming and activating stimuli differ and are specific to disease, tissue and cell type.

Point 7: line 105 – “caspase-1 promotes proptosis” may be beneficial to state through GSDMD cleavage. Typo pyroptosis

Point 8: line 255 - define JAK/STAT (Janus kinase/signal transducers and activators of transcription)

Point 9: line 305-311 – Please elaborate on the findings of HMGB1 in AP studies

Point 10: line 312-320 – highlight that it is a hypothesis that Hsp70 may activate the inflammasome in AP as there is no solid evidence. Expand on the evidence of an increase of HSP70 in AP

Point 11: Line 421- “IL-33 is a major mediator of the disease through TNF-a” – please explain this in a little more detail, how are these two mediators linked.

Point 12: Line 355-356 – there is no mention of the activation of the inflammasome in AP by PAMPs in the main body of text but mentioned in Figure 2. Authors should add this into the main body of text or remove from the diagram if there is no evidence.

Point 13: line 254 – Authors should remove the introduction to this figure.

                Line 354 – no need for introduction to figure 2

Point 14 - Line 427 and 428 – “NLRP-3” please remove - , keep this consistent throughout text

Point 15 - Section 4.2 – title needs to allude to cytokines in AP

Point 16: line 440-441 – Authors should state that this finding is in PBMC’s

Point 17: Authors should refrain from stating “in a study by”, directly describe the findings of the study

Point 18: After section 2 another section on an overview of inflammation in AP in general would be beneficial.  

Point 19: Reference 111 is not cited in the main body of text, please either include this reference or remove.

Point 20: In the reference list reference 4 and 27 are the same- please remove 1 and edit main body of text

Point 21: There is an array of studies investigating NLRP3 inflammasome inhibition in AP however only a few have been highlighted in this study. Please include detail on the other inflammasome inhibitors in AP models, a table summarising and outlining all the inhibitors tested AP models will help with this. Below are examples of 3 studies which have been not been discussed in this review, however there may be more.

Add detail about INF-39 inflammasome inhibitor in section 6 (https://onlinelibrary.wiley.com/doi/full/10.1155/2018/1294951)

Add detail about Cordycepin inhibition of NF-kB and inflammasome in AP (https://www.sciencedirect.com/science/article/abs/pii/S0024320520303933)

Include study on hydrogen rich saline inhibition of NLRP3 inflammasome activation in AP (https://onlinelibrary.wiley.com/doi/full/10.1155/2014/930894)

Author Response

We would like to thank you for taking the time to help us improve our manuscript. Here are our responses to reviewer comments. Changes made in the text based on reviewer comments are highlighted in yellow.

Comment 1: Please add a little bit of information on how AP is currently treated and why these treatments fail.

Response 1: More information on current AP treatment has been added in the "Introduction", and lack of specificity and suppression of the inflammatory process have been highlighted as reasons for treatment failure in severe cases (lines 47-50).

Comment 2: sections 2 and 3.1 are very similar and lots of overlapping material, this could be condensed into 1 section.

Response 2: These sections have been condensed into one section (section 2). Paragraph 3.1. has been deleted. References have been changed accordingly.

Comment 3: In line 146 authors state “The expression level of pro-IL-18 does not seem to be influenced by priming” however in line 244 authors have written that NF-kB promotes transcription of IL-18”, please revisit this.

Response 3: The sentence "The expression levels of ASC, pro-caspase-1, and pro-IL-18 do not seem to be influenced by priming signals" in line 146 has been removed from text.

Comment 4:  line 172 - define calcium (Ca2+)  in first instance 

                      line 177 - define chloride (Cl-) here 

Response 4: Both calcium (Ca2+) (line 199) and chloride (Cl-) (line 203) have been defined accordingly.

Comment 5:  line 211 - 215 - this is repetitive of the previous paragraph and can be removed or condensed.

Response 5: The paragraph mentioned has been removed. The sentence "These pores cause cell swelling and eventually lead to cell lysis" has been changed to " These pores cause cell swelling and eventual disruption of the integrity of the cell membrane, leading to cell lysis." (line 231-233).

Comment 6: section 3.2.1 and section 3.2.2 - I think it is important to note here that inflammasome priming and activating stimuli differ and are specific to disease, tissue and cell type.

Response 6: The sentence "Inflammasome priming and activating stimuli differ and are specific to disease, tissue and cell type" has been added to paragraph 4.1.1 (lines 169-170).

Comment 7: line 105 – “caspase-1 promotes proptosis” may be beneficial to state through GSDMD cleavage. Typo pyroptosis

Response 7: Sentence has been changed to "Through gasdermin D (GSDMD) cleavage, caspase-1 also triggers pyroptosis". Gasdermin D (GSDMD) was defined in first instance (lines 116/117).

Comment 8: line 255 - define JAK/STAT (Janus kinase/signal transducers and activators of transcription).

Response 8: JAK/STAT has been defined at first instance (lines 246-247).

Comment 9: line 305-311 – Please elaborate on the findings of HMGB1 in AP studies.

Response 9: We have provided more details regarding findings of HMGB1 in AP studies (lines 326-335).

Comment 10: line 312-320 – highlight that it is a hypothesis that Hsp70 may activate the inflammasome in AP as there is no solid evidence. Expand on the evidence of an increase of HSP70 in AP.

Response 10: It has been highlighted that a clear connection between Hsp70 and NLRP3 activation has not been established. Moreover, studies indicating increased concentration and protective effects of intracellular Hsp70 in AP have been added (lines 339-350) 

Comment 11: Line 421- “IL-33 is a major mediator of the disease through TNF-a” – please explain this in a little more detail, how are these two mediators linked.

Response 11:  The results of the mentioned study the possible association between IL-33 and TNF-a (lines 458-462).

Comment 12: Line 355-356 – there is no mention of the activation of the inflammasome in AP by PAMPs in the main body of text but mentioned in Figure 2. Authors should add this into the main body of text or remove from the diagram if there is no evidence.

Response 12: Information and references regarding activation of different pathways through PAMPs and PRR interaction has been added, further showing a possible connection of bacterial translocation and NLRP3 activation in AP (lines 370-377). Figure 2 has been modified accordingly.

Comment 13: line 254 – Authors should remove the introduction to this figure.

                Line 354 – no need for introduction to figure 2

Response 13: The introductions to both figures have been removed.

Comment 14: Line 427 and 428 – “NLRP-3” please remove - , keep this consistent throughout text.

Response 14: NLRP3 has been typed in lines 468-469, and kept consistent throughout text.

Comment 15 - Section 4.2 – title needs to allude to cytokines in AP.

Response 15: Title of paragraph 5.3 has been changed to "NLRP3 activation and the resulting effect of cytokines in AP". 

Comment 16: line 440-441 – Authors should state that this finding is in PBMC’s

Response 16: The text in lines 483-484 has been modified to clearly state that this finding is in PBMC’s.

Comment 17: Authors should refrain from stating “in a study by”, directly describe the findings of the study.

Response 17: The phrase "in a study by" has been removed throughout the text, and findings of each study have been directly described.

Comment 18: After section 2 another section on an overview of inflammation in AP in general would be beneficial.

Response 18:  A new section titled "3. The Inflammatory Process in Acute Pancreatitis" has been added, containing information regarding inflammation in AP. To avoid repetition of similar information, the paragraph previously titled "4.1. The inflammatory process in acute pancreatitis" has been renamed to "5.1. NLRP3 Inflammasome is activated during Acute Pancreatitis ", and other studies associated with NLRP3 activation in AP have been cited. The main text and references have been modified accordingly (lines 138-159).

Comment 19: Reference 111 is not cited in the main body of text, please either include this reference or remove.

Response 19: This reference has been cited in the main body of text, but its new number is 130 (line 586).

Comment 20: In the reference list reference 4 and 27 are the same- please remove 1 and edit main body of text.

Response 20: Reference 27 has been removed. The main text has been edited accordingly.

Comment 21: There is an array of studies investigating NLRP3 inflammasome inhibition in AP however only a few have been highlighted in this study. Please include detail on the other inflammasome inhibitors in AP models, a table summarising and outlining all the inhibitors tested AP models will help with this. Below are examples of 3 studies which have been not been discussed in this review, however there may be more.

Response 21: We have added more studies regarding NLRP3 inhibitors in the discussion (references 100, 114, 129) (lines 530-531, 575-581)  , including the 3 suggested in this comment (references 66, 125, 126) (lines 534-536, 564-569) as well as more details about studies already mentioned in the text (lines 543-553, 582-593). Table 2 (line 510) summarizes and outlines all the inhibitors test AP models that are also mentioned in the main text.

Reviewer 2 Report

Comments and Suggestions for Authors

This review highlights the significance of NLRP3 inflammasome as a potential therapeutic target for acute pancreatitis (AP). It underscores the gap between promising animal studies and the lack of clinical trials, emphasizing the urgent need for research in this area. By focusing on the inflammatory processes involved in AP, it sets a clear direction for future investigations that could lead to improved treatment options for patients. This review could ultimately enhance our understanding and management of this serious condition.

1. The author has included extensive information about the NLRP3 inflammasome as a potential therapeutic target for acute pancreatitis (AP). However, discussing potential limitations or contradictory studies could provide a more balanced perspective on inflammasome pathways and enhance reader engagement.

2.  Including a table summarizing studies that support or oppose the role of NLRP3 as a therapeutic target in AP could strengthen the manuscript.

3. In the introduction, presenting other available therapeutic targets before emphasizing the significance of NLRP3 in AP may enhance clarity and contextual understanding.

4. A table detailing preclinical or clinical studies involving NLRP3 inhibitors in the context of AP would further bolster the manuscript.

5. Expanding the discussion with more clinical studies could enhance the comprehensiveness of the article and provide deeper insights.

6. In the conclusion, suggesting specific areas for future research or clinical trials related to NLRP3 in AP treatment could add value.

7. Overall, the author has effectively provided valuable information about NLRP3 in AP, which could be insightful for future research initiatives.

Author Response

We would like to thank you for taking the time to help us improve our manuscript. Here are our responses to reviewer comments. Changes made in the text based on reviewer comments are highlighted in yellow.

Comment 1: The author has included extensive information about the NLRP3 inflammasome as a potential therapeutic target for acute pancreatitis (AP). However, discussing potential limitations or contradictory studies could provide a more balanced perspective on inflammasome pathways and enhance reader engagement.

Response 1: Information regarding possible limitations in the use of NLRP3 inhibitors have been added to the discussion in section 7, especially regarding IL1 antagonists, inhibition of NF-κB signaling and antibiotic treatment (lines 520-524, 594-599, 605-609).

Comment 2: Including a table summarizing studies that support or oppose the role of NLRP3 as a therapeutic target in AP could strengthen the manuscript.

Response 2: Table 1 (line 315) summarizes studies that support the role of NLRP3 as a therapeutic target in AP. Possible adverse effects of NLRP3 inflammasome inhibition are discussed in section 7.

Comment 3: In the introduction, presenting other available therapeutic targets before emphasizing the significance of NLRP3 in AP may enhance clarity and contextual understanding.

Response 3: Other pharmacologic agents examined in clinical trials have been presented in the introduction, with none of them exhibiting a substantial clinical benefit, indicating a possible need for other therapeutic targets, such as NLRP3 (lines 51-55).

Comment 4: A table detailing preclinical or clinical studies involving NLRP3 inhibitors in the context of AP would further bolster the manuscript.

Response 4: Studies involving NLRP3 inhibitors in the context of AP are described in Table 2 (line 510).

Comment 5: Expanding the discussion with more clinical studies could enhance the comprehensiveness of the article and provide deeper insights.

Response 5: We have added many clinical studies to the discussion in different paragraphs including 5.1 (references 67-69, lines 304-313), 5.2.1. (lines 339-350), 5.2.2. (references 82-83, lines 370-377) and section 7 (references 66,100,114,125,126, 129 in lines 530-531, 534-536, 564-569, 575-581) , as well as a new section discussing the inflammatory process in acute pancreatitis (section 3, line 123). 

Comment 6: In the conclusion, suggesting specific areas for future research or clinical trials related to NLRP3 in AP treatment could add value.

Response 6: Specific areas for future research, including optimal dose, specificity and safety profile of NLRP3 inhibitors in the clinical context, have been suggested in the conclusion (lines 624-628).

Comment 7: Overall, the author has effectively provided valuable information about NLRP3 in AP, which could be insightful for future research initiatives.

Response 7: Thank you for your guidance and for helping us improve our manuscript.